REVIEW-SYMPOSIUM

# Changes in cellular Ca²⁺ and Na⁺ regulation during the progression towards heart failure

Kenneth T. MacLeod 🆔

*National Heart & Lung Institute, Imperial Centre for Translational and Experimental Medicine, Imperial College, Hammersmith Hospital, London, UK*

Handling Editors: Laura Bennet & Beth Habecker

The peer review history is available in the Supporting Information section of this article (https://doi.org/10.1113/JP283082#support-information-section).

**Abstract**  In adapting to disease and loss of tissue, the heart shows great phenotypic plasticity that involves changes to its structure, composition and electrophysiology. Together with parallel whole body cardiovascular adaptations, the initial decline in cardiac function resulting from the insult is compensated. However, in the long term, the heart muscle begins to fail and patients with this condition have a very poor prognosis, with many dying from disturbances of rhythm. The surviving myocytes of these hearts gain Na⁺, which is positively inotropic because of alterations to Ca²⁺ fluxes mediated by the Na⁺/Ca²⁺ exchange, but compromises Ca²⁺-dependent energy metabolism in mitochondria. Uptake of Ca²⁺ into the sarcoplasmic reticulum (SR) is reduced because of diminished function of SR Ca²⁺ ATPases. The result of increased Ca²⁺ influx and reduced SR Ca²⁺ uptake is an increase in the diastolic cytosolic Ca²⁺ concentration, which

**Ken MacLeod** was educated at Aberdeen and Edinburgh Universities, then spent some time in Don Bers' laboratory at the University of California, Riverside, CA, USA, funded by an American Heart Association Fellowship. He is now Professor of Cardiac Physiology and Head of the Cardiac Function Section at the National Heart & Lung Institute, Imperial College. His research focuses on the electrical and contractile events in the heart and how these change during the progression of heart disease. Of particular interest at present are the molecular and cellular mechanisms that can cause arrhythmias and the influence of sex hormones on cardiac function.

The Journal of Physiology

promotes spontaneous SR $Ca^{2+}$ release and induces delayed afterdepolarisations. Action potential duration prolongs because of increased late $Na^+$ current and changes in expression and function of other ion channels and transporters increasing the probability of the formation of early after-depolarisations. There is a reduction in T-tubule density and so the normal spatial arrangements required for efficient excitation–contraction coupling are compromised and lead to temporal delays in $Ca^{2+}$ release from the SR. Therefore, the structural and electrophysiological responses that occur to provide compensation do so at the expense of (1) increasing the likelihood of arrhythmogenesis; (2) activating hypertrophic, apoptotic and $Ca^{2+}$ signalling pathways; and (3) decreasing the efficiency of SR $Ca^{2+}$ release.

(Received 16 May 2022; accepted after revision 2 August 2022; first published online 10 August 2022)

**Corresponding author** K. T. MacLeod: National Heart & Lung Institute, Imperial Centre for Translational and Experimental Medicine, Imperial College, Hammersmith Hospital, Du Cane Road, London W12 0NN, UK. Email: k.t.macleod@imperial.ac.uk

**Abstract figure legend** In adapting to the loss of functioning tissue, the heart changes its structure, composition and electrophysiology at whole organ and cell level and the body activates its neurohormonal systems that adjust its vascular function and blood volume. These adaptations provide compensation for the loss of functioning tissue helping to maintain cardiac output but can precipitate sequelae that, over time, become pro-arrhythmic.

## Introduction

Things sweet to taste prove in digestion sour.
William Shakespeare

'Richard II' (1595) act 1, sc. 3, l. 236
John of Gaunt, Duke of Lancaster

The heart has phenotypic plasticity in adapting to disease and loss of tissue. An example is increasing the size of its muscle cells to overcome the loss of working myocardium following an ischaemic episode. This is a hypertrophic response of the remote tissue to the formation of non-functional scar tissue at the site of the infarction. The response mechanisms, dubbed 'plastic rescue' by evolutionary biologists (Snell-Rood et al., 2018), usually enable the organ, and us, to survive the cardiac insult. However, plastic rescue can be troublesome. Changes to the function of cardiac muscle ion channels and proteins involved in $Ca^{2+}$ homeostasis that are initially beneficial to function can become maladaptive, often setting in motion a cascade of events that, in the heart, provides a setting for disorders of rhythm to manifest themselves.

The primary aims of the present review are (1) to illustrate how the plastic heart modifies mechanisms controlling intracellular $Na^+$, $Ca^{2+}$, and $K^+$ to initially support better function and (2) to explain how these changes can become detrimental to function in the longer term. The regulation of these ions is intertwined and forms homeostatic centre-points at which the electrical events work harmoniously with the changes in intracellular ion activities governing contraction and relaxation. When this series of reference points becomes disorganised because of activated stress responses, the result is costly for the organ and organism in the longer term. There will be no attempt to review all that is known about cardiac arrhythmias but, instead, an explanation is provided of our current understanding of the remodelling responses and how they may form a pro-arrhythmic substrate. Secondary aims are to inform a general readership of the changes taking place in cardiac myocytes during the transition to heart failure and to entice enterprising and interested scientists of various disciplines to investigate avenues of the processes that may lead to the development of rational and novel therapies.

## Heart failure (HF)

Following cardiac injury of some sort (usually myocardial infarction stemming from coronary artery disease, but also insults that increase afterload, such as hypertension and valve disease), a series of physiological responses is initiated that involves activation of the sympathetic nervous system (SNS), the renin–angiotensin–aldosterone system (RAAS) and release of natriuretic peptides, together with structural, contractile and electrophysiological changes to the heart itself. The increased contractility and heart rate following SNS activation and retention of salt and water combined with peripheral arterial vasoconstriction resulting from RAAS activation provide initial cardiovascular compensation for a decline in cardiac function (Hartupee & Mann, 2017). However, this neurohormonal response does not support the long-term maintenance of sufficient cardiac output to meet the body's requirements for oxygen and nutrients. Sustained SNS and RAAS activation becomes damaging through (1) the gradual retention of salt and water leading to volume and pressure overload on the heart and (2) changes to arterial and

myocardial stiffness in part triggered by the increase in angiotensin II levels that encourages perivascular and interstitial collagen proliferation. The release of atrial and brain natriuretic peptides increases in response to the volume and pressure overloads and, normally, they act as antagonists to the effects of angiotensin II, reducing vascular tone, aldosterone secretion and renal tubule sodium reabsorption. However, despite an increase in the levels of these peptides, their effects become diminished. Over time, the sustained pressure and/or volume overloads increase protein synthesis and generate a hypertrophic response characterised by cardiac enlargement with rearrangement and growth of the sarcomeres, activation of intracellular signalling cascades, fibrosis development producing areas of scar tissue, progressive loss of parasympathetic tone and complex inflammatory responses. At the cellular level, there is $\beta$-adrenergic receptor desensitisation, together with changes to myocyte biology, structure and mitochondrial energetics. All these systems play a role in the overall 'remodelling' of the heart (Dobaczewski et al., 2011; Heger et al., 2016; Riehle & Bauersachs, 2019) and produce a complex syndrome with considerable system interactions.

Activation of the SNS and RAAS is why $\beta$-adrenergic receptor blockers, angiotensin-converting enzyme inhibitors, angiotensin receptor blockers and aldosterone antagonists are first-line therapeutic interventions. The reduced effects of the natriuretic peptides as heart failure develops have also encouraged the development of neprilysin or neutral endopeptidase inhibitors that reduce the degradation of the peptides.

Irrespective of the myriad of factors triggered in response to the initial insult, when the heart starts to fail, it inexorably follows the path of a chronic, progressive condition with very poor prognosis. Approximately 60% of people diagnosed with HF are dead within 5 years (Groenewegen et al., 2020; McMurray & Stewart, 2000).

Despite a much better understanding of the elements involved in the disease process and more targeted symptomatic treatments, there is no cure and, unfortunately, HF is becoming more common. Although there is a lack of consensus on precise clinical values that determine its diagnosis, its prevalence in the general population is estimated to be between 1% and 3% and it is more common in people aged >60 years. In this group, the prevalence of left ventricular diastolic dysfunction (HF with preserved ejection fraction; HFpEF) ranges from 16% to 53%, and left ventricular systolic dysfunction (HF with reduced ejection fraction; HFrEF) ranges from 3% to 9% (van Riet et al., 2016).

**Arrhythmia in HF.** Although there may be fewer deaths as a result of arrhythmia in patients with HFpEF, about half of those with HF will die from a disturbance of rhythm (Mozaffarian et al., 2007; Vaduganathan et al., 2018).

The question arises: what causes these abnormal rhythms? It is highly probable that the multiplicity of alterations to structure, mechanical and electrical processes play individual parts. Because the concentration of $Ca^{2+}$ ions in the cytosol drives contraction and relaxation, it seems sensible to start with a short review of how this ion is controlled and how this control is modified as the heart fails.

### Ca influx, release, uptake and efflux

In the normal heart, the coupling of electrical excitation to contraction (EC coupling) involves the interaction of a number of cellular proteins concerned with $Ca^{2+}$ homeostasis. $Ca^{2+}$ influx through L-type $Ca^{2+}$ channels (LTCCs) located in the surface membrane (sarcolemma) promotes further release of stored $Ca^{2+}$ from the sarcoplasmic reticulum (SR) via the SR $Ca^{2+}$-release channel (the ryanodine receptor, RyR) by a process known as $Ca^{2+}$-induced $Ca^{2+}$ release (CICR) (Fabiato, 1985). Both fluxes of $Ca^{2+}$ combine to initiate contraction. Two main systems are involved in removing $Ca^{2+}$ from the cytoplasm and so inducing relaxation. $Ca^{2+}$ is pumped back into the SR by the phospholamban-regulated SR $Ca^{2+}$ ATPase (SERCA2a) and extruded from the cell by the sarcolemmal $Na^+/Ca^{2+}$ exchange (NCX) (Bers, 2002, 2008). Although there are species differences, SERCA and NCX contribute $\sim$70% and 25%, respectively, towards relaxation (Bers, 2001). In steady state conditions, the amount of $Ca^{2+}$ leaving the cell is the same as the amount entering it (Bridge et al., 1990) so that precise intracellular $Ca^{2+}$ homeostasis is achieved. The phasic increase and decrease of $Ca^{2+}$ that gives rise to the elements of contraction and relaxation, respectively, is generally termed the '$Ca^{2+}$ transient'.

Various elements of the EC coupling system can be modulated by a variety of signalling molecules (Terrar, 2020) but one that has been shown to have particular relevance in HF is calcium/calmodulin-dependent kinase II (CaMKII). This is a serine/threonine protein kinase expressed in many tissues including the heart. $Ca^{2+}$ binding to calmodulin increases its affinity for the CaMKII binding site and subsequent binding increases the activity of CaMKII (Jiang & Wang, 2020), which then catalyses the phosphorylation of many proteins. This series of interactions provides a subtly responsive and physiologically important link between cytosolic $Ca^{2+}$ concentration and the activity of proteins involved in $Ca^{2+}$ influx, release, uptake and efflux. The functions of the main proteins involved in EC coupling are modulated by CaMKII. It can catalyse the phosphorylation of phospholamban to release its inhibition of SERCA2a, therefore increasing $Ca^{2+}$ uptake into the SR. It can catalyse the phosphorylation of RyR, sensitising these channels to $Ca^{2+}$ and so enhancing SR $Ca^{2+}$ release, and, by catalysing the phosphorylation

of LTCCs, it can promote $Ca^{2+}$-dependent facilitation of the $Ca^{2+}$ current, increasing its size and slowing its inactivation. This enhanced $Ca^{2+}$ influx leads to an increase in SR $Ca^{2+}$ content (Maier & Bers, 2007) (Fig. 1).

CaMKII can also target other ion channels. For example, it can catalyse the phosphorylation of the main isoform of the $Na^+$ channel expressed in the heart ($Na_v1.5$) and this leads to a leftward shift in its voltage dependence of inactivation. This has the effect of slowing the recovery of the channel from inactivation (Takla et al., 2020).

Hypertrophied and failing cardiac tissue displays poorer contraction and slower relaxation (Bing et al., 1971; Gwathmey et al., 1987; Gwathmey et al., 1990; Schouten et al., 1990) than the normal heart. These features are observed in isolated myocytes, implying that at least a portion of the poorer function is a consequence of systems failing at the cellular level (Beuckelmann et al., 1992; Pogwizd et al., 2001; Siri et al., 1991). $Ca^{2+}$ transients are smaller and slower to decay and diastolic $Ca^{2+}$ levels are increased.

The underlying reasons for the changes to $Ca^{2+}$ regulation are now better understood and involve (1) the loss of transverse (t) tubules (Dibb et al., 2009; Lyon et al., 2009; Oyehaug et al., 2013; Song et al., 2005) that provide a structural framework for the close apposition of sarcolemmal LTCCs with the RyR clusters in the SR membrane and produce a microarchitecture vital for the effective conversion of excitation to contraction (synchronous EC coupling); (2) post-translational modifications to RyRs that change their function (Benitah et al., 2021; Houser, 2014b) resulting in more $Ca^{2+}$ leak from the SR and loss of RyRs that reduce release (Lachnit et al., 1994; Milnes & MacLeod, 2001; Vatner et al., 1994), which, in combination, lead to a reduction in SR $Ca^{2+}$ content and less efficacious release; (3) reduced function of SR $Ca^{2+}$ ATPase (i.e. SERCA2a) protein that results in slower and reduced $Ca^{2+}$ reuptake into the SR (Arai et al., 1993, 1994; de la Bastie et al., 1990; Hasenfuss et al., 1994; Kiss et al., 1995); and (4) increased expression of NCX that alters the competition between $Ca^{2+}$ uptake into the SR and $Ca^{2+}$ efflux (Hobai & O'Rourke, 2000; Reinecke et al., 1996; Studer et al., 1997).

Although the release, uptake and efflux of $Ca^{2+}$ are compromised in HF, the trigger for release (i.e. $Ca^{2+}$ influx via LTCCs) remains, although this may change in size. Some studies report that peak values of the current measured in cells from failing hearts are no different compared to control myocytes (Beuckelmann et al., 1991; Mewes & Ravens, 1994), whereas others report that the current is reduced (Terracciano et al., 2003) or dependent on phosphorylation (Chen et al., 2002).

**T-tubules.** Although there is debate about the size of the $Ca^{2+}$ trigger changing in HF, the efficiency with which it stimulates release from the SR may be compromised because of disruption to the t-tubule network. T-tubule disruption is becoming a more common observation in a variety of cardiac pathologies. Transverse tubules

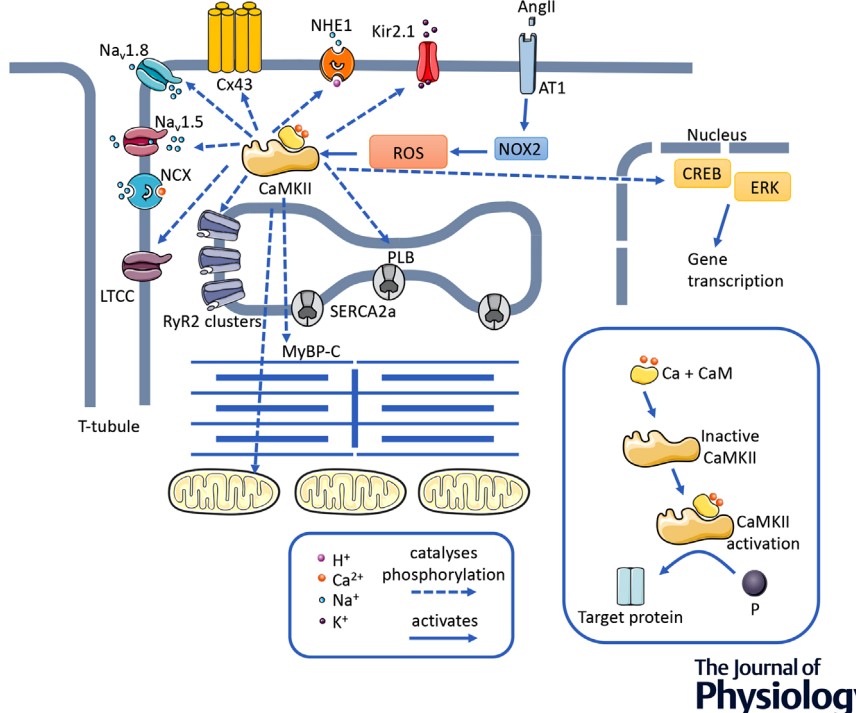

**Figure 1. Signalling pathways and effector proteins modulated by CaMKII**
Showing some of the signalling pathways and effector proteins modulated by CaMKII. The inset illustrates the steps leading to CaMKII activation.

(i.e. t-tubules) are invaginations of the cardiac myocyte surface membrane that penetrate the cytoplasm and form a network by connecting with longitudinal (axial) tubules. Many ion channels and receptors are located in the t-tubules but they have particular importance in ensuring the optimum microarchitecture is obtained between LTCCs (in the surface membrane) and the SR $Ca^{2+}$ release channels (RyR) (in the underlying SR membrane). The t-tubules allow LTCCs to be closely positioned to a cluster of RyRs so that synchronous SR release can be controlled by the incoming $Ca^{2+}$ flux (Fig. 2). Optimum spacing and positioning is vital to the efficacy and stability of the CICR process. HF is characterised by a reduction in the number and organisation of t-tubules in parts of the cells and this leads to delays in $Ca^{2+}$ release in these areas resulting in poor synchronisation of RyR firing, with many clusters being activated later by $Ca^{2+}$ released from their neighbours (Litwin et al., 2000; Louch et al., 2006). The result is a slowing in the rising phase of the $Ca^{2+}$ transient. The disorganisation of cell structure that coincides with the remodelling processes leaves some parts of the cells with a broken signalling link between LTCCs and RyRs, such that some clusters of RyRs are stranded without the trigger command and cannot respond synchronously.

Stranded clusters or unclustered RyRs (termed orphaned and rogue RyRs, respectively) (Sobie et al., 2006; Song et al., 2006) probably gate differently in response to local changes in $Ca^{2+}$ concentration, particularly if they undergo phosphorylation mediated through PKA and CaMKII pathways that occur because of the breakdown in local signalling as a result of t-tubule and caveolae disorganisation. Co-operativity or coupled gating between clusters also appears to decrease, so encouraging less synchronous and more disorganised RyR firing. The modelling work by Lu et al. (2010) suggests that $Ca^{2+}$ concentrations in the vicinity of orphaned RyR clusters and rogue RyRs are poorly controlled with the result that orphaned clusters may activate, thereby increasing the local $Ca^{2+}$ concentration that nearby rogue RyRs augment. The rogue RyRs increase the opening probability of neighbouring clusters allowing $Ca^{2+}$ sparks to coalesce and improve the chances of $Ca^{2+}$ wave formation and propagation (Chen et al., 2018).

**Modifications to RyRs.** The failure of the SR to release $Ca^{2+}$ may effectively be a result not only of variations in RyR location, but also changes in their density or the ratio of the receptors to LTCCs (Lachnit et al., 1994; Milnes & MacLeod, 2001; Vatner et al., 1994). Furthermore, post-translational modifications to RyRs in the failing heart may cause them to behave abnormally. There has been controversy over the nature of these modifications (Houser, 2014a; Benitah et al., 2021) that continues (Alvarado & Valdivia 2020; Dridi et al. 2020). However, many studies have allowed a consensus to be built with respect to CaMKII-mediated phosphorylation (Fischer et al., 2013) and decreased *S*-nitrosylation (Gonzalez et al.,

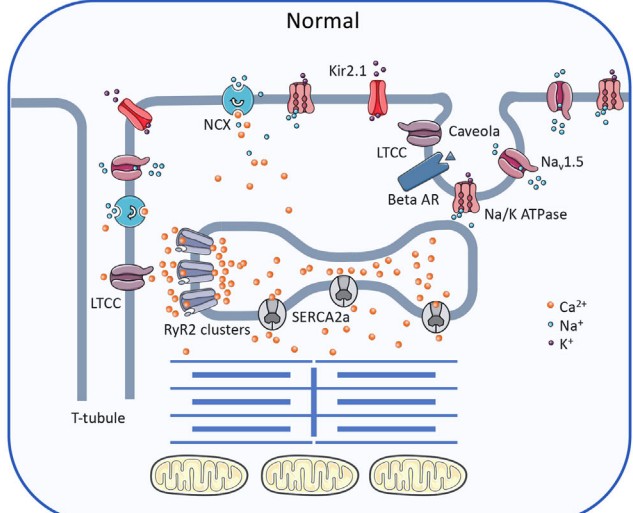
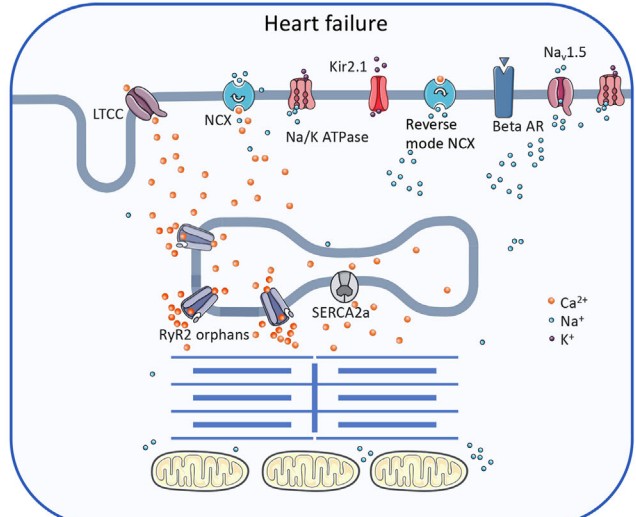

**Figure 2. Comparison of Ca²⁺ and Na⁺ regulation in the normal and failing cardiac myocyte**
Key differences are: (1) the loss of transverse (t) tubules and caveoli breaking signalling links importantly between L-type Ca²⁺ channels (LTCCs) and ryanodine receptors (RyRs); (2) post-translational modifications to RyRs that result in more Ca²⁺ leak from the SR; (3) reduced function of SR Ca²⁺ ATPase (SERCA2a); (4) increased expression of Na⁺/Ca²⁺ exchange (NCX); (5) an increase in intracellular Na⁺ concentration altering mitochondrial function and increasing reverse mode NCX.

2007) of RyRs increasing the channel open time, leading to diastolic $Ca^{2+}$ leak in HF (Figs 1 and 2)

**Caveolae.** In addition to t-tubules, there are smaller invaginations of the surface membranes of cardiac myocytes of between 50 and 100 nm in diameter known as caveolae. The protein responsible for their formation is caveolin, which co-ordinates the membrane lipid to form small pouches that have extracellular access. Caveolae essentially act as domains that gather together certain proteins involved in initiating signalling cascades, allowing spatially-confined interactions. Parts of the beta adrenergic signalling system are located in caveolae, as are some ion transporters ($Na^+$, $Ca^{2+}$ and $K^+$ channels and the $Na^+/K^+$ ATPase) (Gorelik et al., 2013; Gratton et al., 2004). Caveolae help maintain normal cell physiology and are probably involved in stress responses (Schilling et al., 2018) and EC coupling in the heart (Calaghan & White, 2006).

There is evidence in HF that the expression of caveolin-3 protein and its coding mRNA decrease, with the amount of decrease closely related to the degree of left ventricular dysfunction (Feiner et al., 2011). Correspondingly, transgenic mice with cardiac-specific overexpression of caveolin-3 that are subjected to transverse aortic constriction have better cardiac function, less hypertrophy and overall better survival than control mice with the same aortic insult (Horikawa et al., 2011).

**Reduced function of SERCA.** Not only is there reduced efficiency of the CICR process, but also there is less $Ca^{2+}$ in the SR to be released. SR $Ca^{2+}$ content is decreased in HF. Defective operation of the RyRs in the form of increased leak from the receptors can partially account for the lower SR $Ca^{2+}$ content in diseased hearts (Litwin et al., 2000; Shannon et al., 2003). The other mechanism that is defective and can explain the decreased SR $Ca^{2+}$ content is the reduced function of SERCA2a. Many studies find that failing hearts have lower levels of SERCA2a protein and its mRNA (Arai et al., 1994; de la Bastie et al., 1990; Hasenfuss et al., 1994; Matsui et al., 1995; Qi et al., 1997), although attention has also been directed to changes in the way that the protein is regulated.

SERCA2a function is regulated by phospholamban that normally exerts a tonic inhibition on the molecule. When phosphorylated, either by cAMP-dependent or $Ca^{2+}$/calmodulin-dependent protein kinase, the inhibitory action of phospholamban is relieved. This type of action allows SERCA2a activity to be modulated depending on the activities of these kinases and, for example, allows it to play a key role in loading the SR with $Ca^{2+}$ to produce the positive inotropic and lusitropic effects of beta adeno-receptor agonists. Most studies find that the amount of phospholamban is not altered

in HF (MacLennan & Kranias, 2003) but, because of the decrease in the amounts of SERCA2a, there is a relative increase in the ratio of phospholamban to SERCA2a and this would cause increased inhibition of the SR $Ca^{2+}$ ATPase, reducing SR $Ca^{2+}$ content and prolonging the declining phase of the $Ca^{2+}$ transient.

Of note, rescue of function has been achieved by adenoviral gene transfer of SERCA2a into isolated cardiac myocytes from failing human hearts and into whole hearts *in vivo* (del Monte et al., 1999; Miyamoto et al., 2000). The $Ca^{2+}$ transients and contraction profiles were restored in the former study and there was significant improvement in left ventricular function in the latter. Modifying cellular $Ca^{2+}$ regulation by overexpressing SERCA2a also reduces ventricular arrhythmias (Prunier et al., 2008) and improves haemodynamic, echocardiographic and molecular biology assessments of cardiac function (Byrne et al., 2008).

**$Na^+/Ca^{2+}$ exchange.** Reduced SR $Ca^{2+}$ content can be caused not only by decreased SERCA2a activity or expression or increased leak though RyRs, but also by increased expression of the NCX because the two transporters compete for intracellular $Ca^{2+}$ at the same time as bringing about relaxation. Usually $Na^+$ influx is coupled to the efflux of $Ca^{2+}$ (forward mode of the exchange) but the direction of ion movement is dependent on membrane potential, as well as the extracellular and intracellular concentrations of $Na^+$ and $Ca^{2+}$. The potential at which ion movement switches direction (reverse mode) is called the reversal potential. The reversal potential is readily encountered under physiological conditions and so factors that influence intracellular $Na^+$ concentration will ultimately affect the intracellular $Ca^{2+}$ concentration and consequently twitch and passive (tonic) force production, in turn, determinants of cardiac output and ventricular filling. The intracellular $Na^+$ concentration increases in HF and the action potential prolongs and so overexpression of the NCX protein on the one hand may aid $Ca^{2+}$ efflux but, on the other, may support more $Ca^{2+}$ influx, adding to the increased diastolic $Ca^{2+}$ concentration caused by poorer SERCA2a uptake and enhanced RyR leak (Fig. 3). Evidence of increased expression of the NCX in HF is inconsistent (Sipido et al., 2002), although several studies do report enhanced amounts of transporter protein (Hasenfuss et al., 1999; Hobai & O'Rourke, 2000; O'Rourke et al., 1999; Studer et al., 1994).

## Intracellular $Na^+$ homeostasis

The concentration of intracellular $Na^+$ in cardiac myocytes is determined by the leak of the ion into the cells and the expulsion of the ion from the cells: the pump/leak

balance. The $Na^+/K^+$-ATPase (or $Na^+/K^+$ pump) has two main functions, conserving both (1) the transmembrane $K^+$ ion concentration difference upon which the diastolic membrane potential largely depends and (2) the transmembrane $Na^+$ ion concentration difference, so enabling its influx through activated channels and sustaining transport processes that couple its movement with other ions, amino acids and metabolites.

Inhibiting the $Na^+/K^+$ pump increases the intracellular $Na^+$ concentration (Deitmer & Ellis, 1977, 1978; Eisner, 1990; Eisner & Lederer, 1980a, 1980b) and alters the balance of $Ca^{2+}$ flux generated by the NCX during the entire cardiac cycle, both systole and diastole (Bers, 1987; Bennett et al., 1999). The time spent in reverse mode increases, thus augmenting $Ca^{2+}$ influx and reducing exchange-mediated $Ca^{2+}$ efflux. A portion of the augmented influx is taken up into the SR (Bennett et al., 1999) by SERCA2a, increasing SR $Ca^{2+}$ load so that more $Ca^{2+}$ is available for release at the next beat, thereby strengthening the next and subsequent contractions. In this way, inhibitors of the pump, notably the cardiac glycosides, have a positive inotropic action on ventricular muscle. Indeed, glycosides such as digoxin were the earliest treatments for the failing heart, following the work of Withering in 1785, who described the advantages of administering extracts of the foxglove, *Digitalis purpurea*, to patients with HF, particularly if their heart rhythm was irregular (Hauptman & Kelly, 1999). Two hundred and thirty-seven years later, digoxin is still

in clinical use and reduces hospitalisations in patients with chronic HF, particularly when combined with $\beta$-blockers (Gheorghiade et al., 2006). However, the improved contractility offered by $Na^+/K^+$ pump inhibition comes at a price, particularly in the vast majority of patients with coronary atheroma who are admitted to hospital with poor ventricular function following acute myocardial infarction. In this group of patients, digoxin use is associated with an increased risk of sudden death (Bigger et al., 1985; Spargias et al., 1999). Digoxin also increases vagal activity by acting on central and peripheral components of the parasympathetic nervous system (Watanabe, 1985). This reduces atrioventricular node conduction and shortens atrial refractory periods, rendering the atria more susceptible to atrial fibrillation.

Although $Na^+/K^+$ pump inhibition improves contractility, the increase in intracellular $Na^+$ concentration can establish a pro-arrhythmic substrate that develops in several ways. Because cellular $Ca^{2+}$ efflux over the cardiac cycle is compromised, both the cytoplasmic $Ca^{2+}$ concentration and SR $Ca^{2+}$ content increase. These conditions can promote spontaneous release of $Ca^{2+}$ from the SR often in the form of $Ca^{2+}$ waves that activate the NCX and, in some species, stimulate $Ca^{2+}$-activated $Cl^-$ current (Trafford et al., 1995, 1998; Wier et al., 1987). $Ca^{2+}$ waves also cause cell alternans (where the amplitude of the $Ca^{2+}$ transient alternates out of phase in different regions of the same cell) that trigger sudden repolarisation changes affecting action potential duration

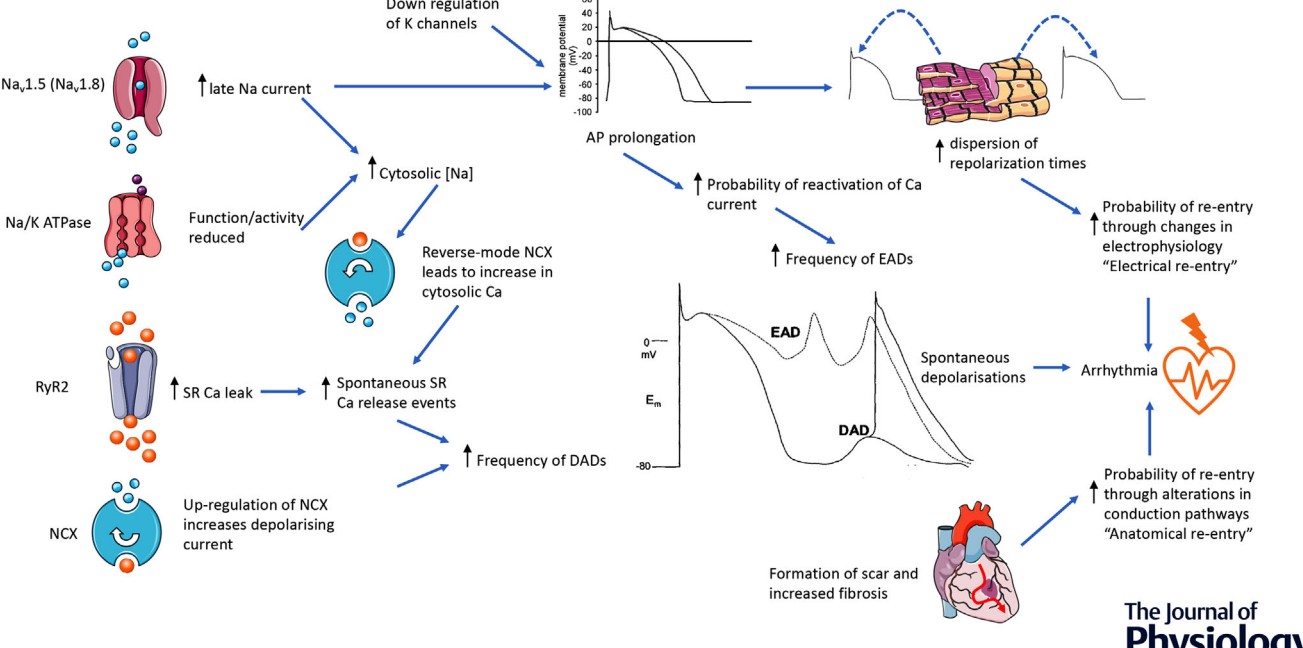

**Figure 3. Changes to ion homeostatic mechanisms and formation of scar tissue**
How changes to the function of ion homeostatic mechanisms and formation of scar tissue may provoke proarrhythmic cardiac electrophysiology.

(Xie & Weiss, 2009; Xie et al., 2010). In both cases, early and delayed afterdepolarisations (EADs and DADs) form and, if they are sufficiently large, membrane potential may reach a voltage range that activates $Na^+$ channels and a premature upstroke may ensue (Schlotthauer & Bers, 2000) (Fig. 3).

If repolarising currents are reduced, as occurs in HF (see below), the action potential duration increases and the diastolic membrane potential may become less stable and so the likelihood of DADs initiating larger premature excitatory events increases (Pogwizd et al., 2001).

**The $Na^+/K^+$ pump in HF.** Most studies measuring the intracellular $Na^+$ concentration find that it increases in cardiac hypertrophy (Grey et al., 2001; Jelicks & Siri, 1995) and HF (Baartscheer et al., 2003; Despa et al., 2002; Pieske et al., 2002; Schillinger et al., 2006). The increases in intracellular $Na^+$ may be the result of an increase in $Na^+$ leak into the cells or a reduction in $Na^+/K^+$ pump function or both (Baartscheer et al., 2003; Despa et al., 2002; Semb et al., 1998; Verdonck et al., 2003) (Ke et al., 2020; N¢rgaard et al., 1988; Schwinger et al., 1999). The reduction in pump function may be a result of decreased expression levels of the alpha subunits (the main catalytic sites) of the ATPase, relative changes in the various forms of alpha subunit expression (isoform switches) and/or alterations in the pump activity, but experimental work aimed at differentiating between these possibilities gives an inconsistent picture of their relative importance. One part of the pump protein that has received more recent attention is its regulatory subunit, phospholemman (PLM), which belongs to the FXYD protein family (*FXYD1*). Unphosphorylated PLM tonically inhibits the $Na^+/K^+$ ATPase by reducing its affinity for $Na^+$. The inhibition is removed when it is phosphorylated by protein kinases A and C and imparted on dephosphorylation by protein phosphatases (particularly PP-1 and PP-2A) (Fuller et al., 2013; Fuller et al., 2004). In this way, the activity of the $Na^+/K^+$ ATPase, and therefore the cytosolic $Na^+$ concentration, is regulated by the balance of phosphorylation and dephosphorylation events.

What happens in HF to this regulatory mechanism is not clear. There is evidence for increased PLM protein levels following infarction in rat hearts that results in reduced pump activity (Mirza et al., 2012). Cardiac PLM hypophosphorylation has been noted following aortic constriction in the mouse and can account for the observed decline in $Na^+/K^+$ ATPase current. In the same study, knock-in mice, in which PLM cannot be phosphorylated, given the same amount of aortic constriction, have poorer cardiac function and greater inhibition of pump current compared to wild-type mice (Boguslavskyi et al., 2014). PLM phosphorylation at Ser-68 is decreased in failing human hearts as a result of dilated cardiomyopathy (El-Armouche et al., 2011). These observations generally point to more PLM being unphosphorylated in HF so that the ratio of unphosphorylated PLM to $Na^+/K^+$ ATPase increases and pump activity decreases. However, in a rabbit model of HF, a decrease in both $Na^+/K^+$ ATPase and PLM expression was found but with PLM expression showing the greater reduction. The result was that the PLM to $Na^+/K^+$ ATPase ratio decreases and it was suggested that this rescued possible pump inhibition (Bossuyt et al., 2005).

An alternative, but not mutually exclusive, explanation for reduced function the $Na^+/K^+$ ATPase is linked to the location of its beta subunit. The beta subunit of the $Na^+/K^+$ ATPase is needed to form fully functioning $Na^+/K^+$ pumps. Evidence suggests that this subunit is confined almost entirely to caveolae (Liu & Askari, 2006) implying that functioning pumps are also located in these cell microdomains. Caveolin-3 levels are reduced in failing hearts (Feiner et al., 2011) and this loss will disrupt caveolae structure and probably reduce the number of these important signalling microdomains with abnormal function of the $Na^+/K^+$ ATPases as a possible consequence. Although the regulatory mechanisms of the $Na^+/K^+$ ATPase in HF are still to be clarified, the consensus is that there is decreased activity of the pump and increased intracellular $Na^+$ concentration.

**Na influx in HF.** The main routes of $Na^+$ entry into cardiac myocytes are $Na^+$ channels (mainly $Na_v1.5$), Na/Ca exchange and the two transporter systems that play a role in intracellular pH regulation, the $Na^+/H^+$ exchange and $Na^+/HCO_3^-$ symport. Other transporters such as the $Na^+/Mg^{2+}$ exchanger (Handy et al., 1996; Tashiro et al., 2014) and the $Na^+/K^+/2Cl^-$ cotransporter (Anderson et al., 1996) may mediate some $Na^+$ influx but the sizes of their contributions are uncertain. There is good evidence for there being a larger TTX-sensitive $Na^+$ influx occurring in HF (Despa et al., 2001, 2002) and a probable route may be increased late $Na^+$ current (Maltsev & Undrovinas, 2008; Valdivia et al., 2005). This $Na^+$ influx is associated with action potential prolongation, which is a widely found feature of myocytes isolated from failing hearts. It is becoming apparent that the processes that lead to an increase in late $Na^+$ current are complex but they are gradually being unravelled.

There is now increasing evidence that the modulatory role of CaMKII can become overactive in HF and this can have deleterious consequences for intracellular $Na^+$ (Grandi & Herren, 2014) and $Ca^{2+}$ regulation (Mattiazzi et al., 2015) and be proarrhythmic (Hund & Mohler, 2015). The expression and activity of CaMKII is increased in animal models of HF and in failing human hearts (Hoch et al., 1999; Kirchhefer et al., 1999; Zhang et al., 2003). The overactivity appears to be a result of enhanced

autophosphorylation or oxidation arising as a consequence of an increase in reactive oxygen species (Swaminathan et al., 2012). The effect of enhanced CaMKII activity on Ca$^{2+}$ handling proteins has been described earlier but equally significant are the findings indicating that CaMKII augments late Na$^+$ current and so may promote the increase in intracellular Na$^+$ concentration (Wagner et al., 2006). A fascinating development of this investigative focus on late Na$^+$ current and CaMKII is the observation that CaMKIIδc interacts with the neuronal form of the Na$^+$ channel (Na$_v$1.8) normally expressed in human ventricular cardiomyocytes at low levels but upregulated several fold in HF (Dybkova et al., 2018). In healthy myocytes, Na$_v$1.8 appears to have negligible influence on depolarisation but contributes significantly to action potential prolongation in failing cells (Dybkova et al., 2018). Recent work using cardiac myocytes isolated from patients with HF shows that the increase in Na$^+$ influx via Na$_v$1.8 is dependent on CaMKIIδc (Bengel et al., 2021) (Figs 2 and 3).

Sodium-glucose co-transporter-2 (SGLT2) inhibitors (the gliflozin class of drugs) are used for treating type 2 diabetes. They inhibit the reabsorption of glucose in the renal tubule and so increase its excretion. Evidence obtained from a number of cardiovascular trials (Anker et al., 2021; Fitchett et al., 2019) demonstrates that SGLT2 inhibitors reduce the incidence of HF in patients with diabetes by ∼40% and also indicates that the drugs have significant cardioprotective effects in the absence of diabetes. This suggests they have additional actions independent of glucose control but the mechanisms at play are very unclear, although some have parallels with neurohormonal antagonists (Packer, 2020). Some groups have obtained evidence that the drugs may directly reduce the intracellular Na$^+$ concentration in cardiac myocytes by inhibiting the Na$^+$/H$^+$ exchange (Zuurbier et al., 2021), although this is controversial (Chung et al., 2021), whereas other studies provide evidence that they inhibit the late Na$^+$ current (Philippaert et al., 2021). Therefore, although their underlying mechanism of action remains unresolved, the drugs appear to link cardioprotection in diabetic and non-diabetic patients with aspects of Na$^+$ regulation (Fig. 2).

### Action potential duration

The most consistent electrophysiological finding in cardiac hypertrophy and HF is a prolongation of the ventricular cell action potential (Ahmmed et al., 2000; Beuckelmann et al., 1993; Kaab et al., 1996; Rose et al., 2005; Pogwizd & Bers, 2004; Pogwizd et al., 2001) caused by a change in the expression and function of ion channels and transporters. The ionic currents most consistently shown to change are the main repolarising K$^+$ currents

($I_{Kr}$, $I_{Ks}$, $I_{to}$ and $I_{K1}$) that decrease in density (Ahmmed et al., 2000; Li et al., 2004; Pogwizd & Bers, 2004; Pogwizd et al., 2001; Rose et al., 2005) and the late Na$^+$ current that increases in density (Maltsev & Undrovinas, 2008; Valdivia et al., 2005).

The decrease in K$^+$ currents will lead to a prolongation of the repolarising processes and less stable diastolic membrane potentials, which will present a more arrhythmogenic substrate because the chances of EADs and DADs initiating depolarisations that reach threshold are increased. Action potential prolongation normally increases SR Ca$^{2+}$ content, although this will only happen if SERCA2a Ca$^{2+}$ uptake into the SR is not compromised (Terracciano et al., 1997). If SERCA2a cannot function optimally, then the increased Ca$^{2+}$ influx will cause an increase in diastolic Ca$^{2+}$ concentration that could induce activation of RyR clusters and rogue RyRs leading to unsynchronised SR Ca$^{2+}$ release and DAD formation. There could be a teleological argument made to suggest that action potential prolongation provides compensatory inotropy but this is at the expense of increasing the likelihood of arrhythmogenesis (Pogwizd et al., 1999; Pogwizd et al., 2001) and activation of hypertrophic and apoptotic signalling (Molkentin, 2004; Sapia et al., 2010). Caveolae may also play a role in the electrophysiological changes. The inward rectifying K$^+$ channel K$_{ir2}$ that carries $I_{K1}$ appears to co-localise with caveolin-3 and mutations in caveolin-3 cause an increase in late Na$^+$ current (Vaidyanathan et al., 2018) and so any disruption of caveolae may precipitate changes to depolarisation and repolarisation currents (Fig. 3).

### Na$^+$ and mitochondrial function

The potential aggravating effects of increased intracellular Na$^+$ concentration are not limited to sarcolemmal ion movements. Studies have demonstrated that such increases affect mitochondrial function (Iwai, Tanonaka, Inoue, Kasahara, Kamo et al., 2002; Iwai, Tanonaka, Inoue, Kasahara, Motegi et al., 2002; Kohlhaas et al., 2010; Maack et al., 2006). Increases and decreases in myocyte cytoplasmic Ca$^{2+}$ are mirrored in changes of mitochondrial matrix Ca$^{2+}$ concentration. Increases in matrix Ca$^{2+}$ concentration activate dehydrogenases and phosphorylation enzymes and match cell respiratory capacity and oxidative phosphorylation to energy requirements on a beat-to-beat basis. Mitochondrial Ca$^{2+}$ homeostasis is a balance of Ca$^{2+}$ influx through the mitochondrial uniporter and efflux through the mitochondrial NCX. An increase in cytoplasmic (extramitochondrial) Na$^+$ concentration causes a decrease in mitochondrial matrix Ca$^{2+}$ concentration and reduces oxidative phosphorylation during periods of increased work. Whether this mechanism causes

metabolic changes in HF remains to be shown definitively and the heart may be sufficiently adept to remodel its metabolism to oppose the effects of reductions in oxidative phosphorylation (Aksentijević & Shattock, 2021). Even if ATP delivery is not compromised, increases in cytosolic $Na^+$ concentration lead to net oxidation of NADPH and increases in intracellular reactive oxygen species (ROS) that contribute to cell damage (Bay et al., 2013; Bertero & Maack, 2018; Kohlhaas et al., 2010; Liu et al., 2010; Maack et al., 2006). There is evidence that inhibiting the mitochondrial NCX maintains cardiac function, slowing hypertrophic remodelling and lessening mortality from sudden cardiac death (Liu et al., 2014).

Two other aspects of mitochondrial $Ca^{2+}$ regulation should be noted here. Although mitochondrial $Ca^{2+}$ concentration is important in modifying ATP supply, high concentrations are known to cause mitochondrial swelling and dysfunction. As described earlier, RyR-mediated SR $Ca^{2+}$ leak increases in HF (Litwin et al., 2000; Marx et al., 2000; Shannon et al., 2003) and some findings suggest that these leaks are responsible for pathologically-elevated mitochondrial $Ca^{2+}$ and excessive ROS production (Santulli et al., 2015).

Second, there are a converse series of processes originating from chronic neurohormonal activation that results in persistently raised angiotensin II levels. These increased levels encourage unregulated ROS production (Zablocki & Sadoshima, 2013) that stems from stimulation of NADPH oxidase (Li et al., 2002). The increased ROS levels damage $Ca^{2+}$ handling proteins and signalling molecules and open the mitochondrial permeability transition pores that stimulate apoptotic pathways. Hence, mitochondrial $Ca^{2+}$ is linked to cell life support and cell death.

## Conclusions

It is difficult to determine which of the many physiological mechanisms disturbed by HF precipitates the other events, and so pinpointing potential targets for therapies is not straightforward. It is doubtful that there is a single initiating factor because the heart has a spectrum of adaptations that it invokes to boost function. Most HF is preceded by a period during which function, although perhaps still impaired, is reasonably sustained and provides compensation for the initial insult. However, there are some results that signpost avenues for possible future investigation and therapeutic targeting. First, it has been demonstrated that SERCA2a gene transfer into myocytes from failing hearts increased SR $Ca^{2+}$ content and reduced $Ca^{2+}$ spark-mediated SR leak thereby preventing arrhythmogenic DADs (Lyon et al., 2011). Surprisingly, re-introducing SERCA2a to failing cells also partially restored the t-tubule network, improved the synchronisation of SR $Ca^{2+}$ release and

redistributed $\beta$-adrenoreceptors leading to more effective signalling (Lyon et al., 2012), suggesting that some reverse remodelling may be possible through improvements in $Ca^{2+}$ regulation. Second, a pervasive feature of HF is the increased expression and activity of CaMKII, which promotes many deleterious changes in $Ca^{2+}$ regulation. CaMKII inhibitors have the potential to have therapeutic effects in HF also by improving $Ca^{2+}$ homeostasis. Third, the increase in intracellular $Na^+$ concentration appears to be critical in modulating a number of processes involved in inotropic compensation but it clearly plays an important part in arrhythmogenesis: plasticity coming at a price. It is pertinent to note that angiotensin-converting enzyme inhibitors, drugs with well-established benefit in the treatment of HF, stimulate the $Na^+/K^+$ ATPase by an unknown mechanism (Hool et al., 1995) and this leads to a lower intracellular $Na^+$ concentration. Finding ways to reduce intracellular the $Na^+$ concentration is a therapeutic route worth considering (Shattock, 2009) and a new target could be a reduction in late $Na^+$ current achieved through inhibition of the cardiac disease dependent $Na_v1.8$. In this review, the emphasis has been on global cytosolic changes in $Na^+$, although local ion gradients and control pathways doubtless trigger disturbances in cellular ionic homeostasis. This highlights the importance of structural alterations to t-tubules and caveolae with the intertwined nature of $Na^+$, $Ca^{2+}$ and $K^+$ regulation that was indicated in the Introduction to this review.

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

## Additional information

### Competing interests

The author declares that he has no competing interests.

### Author contributions

K.M. was responsibe for the conception or design of the work, as well as drafting the work or revising it critically for important intellectual content. K.M. approved the final version

of the manuscript submitted for publication. KM agrees to be accountable for all aspects of the work.

## Funding

This work was funded by the British Heart Foundation (BHF) N/A.

## Acknowledgements

Kenneth T. MacLeod is supported by the British Heart Foundation

## Keywords

calcium influx, heart failure, Na$^+$/Ca$^{2+}$ exchange, Na$^+$/K$^+$ pump, sodium homeostasis

## Supporting information

Additional supporting information can be found online in the Supporting Information section at the end of the HTML view of the article. Supporting information files available:

**Peer Review History**

