## [Peer Review History · The Journal of Physiology]

Changes in cellular Ca²⁺ and Na⁺ regulation during the progression towards heart failure

Kenneth T. MacLeod
DOI: 10.1113/JP283082

Corresponding author(s): Kenneth MacLeod (k.t.macleod@imperial.ac.uk)

Review Timeline:

Submission Date:	16-May-2022
Editorial Decision:	07-Jun-2022
Revision Received:	18-Jul-2022
Accepted:	02-Aug-2022

Senior Editor: Laura Bennet

Reviewing Editor: Beth Habecker

Transaction Report:

Dear Dr MacLeod,

Re: JP-SR-2022-283082 "Changes in cellular Ca²⁺ and Na⁺ regulation during the progression towards heart failure" by Kenneth T. MacLeod

Thank you for submitting your invited Review-Symposium to The Journal of Physiology. It has been assessed by a Reviewing Editor and by 2 expert referees and I am pleased to tell you that it is considered to be acceptable for publication following satisfactory revision.

The reports are copied at the end of this email. Please address all of the points and incorporate all requested revisions, or explain in your Response to Referees why a change has not been made.

NEW POLICY: In order to improve the transparency of its peer review process The Journal of Physiology publishes online as supporting information the peer review history of all articles accepted for publication. Readers will have access to decision letters, including all Editors' comments and referee reports, for each version of the manuscript and any author responses to peer review comments. Referees can decide whether or not they wish to be named on the peer review history document.

I hope you will find the comments helpful and have no difficulty in revising your manuscript within 4 weeks.

Your revised manuscript should be submitted online using the links in Author Tasks Link Not Available. This link is to the Corresponding Author's own account, if this will cause any problems when submitting the revised version please contact us.

The image files from the previous version are retained on the system. Please ensure you replace or remove any files that have been revised. Your revised submission should include:

- A Word file of the complete text (including figure legends any Tables);
- An Abstract Figure (with legend in the Article file)
- Each figure as a separate, high quality, file;
- A full Response to Referees;
- A copy of the manuscript with the changes highlighted.
- Author profile. A short biography (no more than 100 words for one author or 150 words in total for two authors) and a portrait photograph of the two leading authors on the paper. These should be uploaded, clearly labelled, with the manuscript submission. Any standard image format for the photograph is acceptable, but the resolution should be at least 300 dpi and preferably more.

- A 'Cover Art' file for consideration as the Issue's cover image;
- Appropriate Supporting Information (Video, audio or data set https://jp.msubmit.net/cgi-bin/main.plex?form_type=display_requirements#supp).

To create your 'Response to Referees' copy all the reports, including any comments from the Reviewing Editor into a Word, or similar, file and respond to each point in colour or CAPITALS and upload this when you submit your revision.

I look forward to receiving your revised submission.

If you have any queries please reply to this email and staff will be happy to assist.

Yours sincerely,

Professor Laura Bennet
Senior Editor
The Journal of Physiology
<https://jp.msubmit.net>
<http://jp.physoc.org>
The Physiological Society
Hodgkin Huxley House
30 Farringdon Lane
London, EC1R 3AW
UK
<http://www.physoc.org>
<http://journals.physoc.org>

REQUIRED ITEMS:

-Please include an Abstract Figure. The Abstract Figure is a piece of artwork designed to give readers an immediate understanding of the Review Article and should summarise the main conclusions. If possible, the image should be easily 'readable' from left to right or top to bottom. It should show the physiological relevance of the Review so readers can assess the importance and content of the article. Abstract Figures should not merely recapitulate other figures in the Review. Please try to keep the diagram as simple as possible and without superfluous information that may distract from the main conclusion of the Review. Abstract Figures must be provided by authors no later than the revised manuscript stage and should be uploaded as a separate file during online submission labelled as File Type 'Abstract Figure'. Please ensure that you include the figure legend in the main article file. All Abstract Figures will be sent to a professional illustrator for redrawing and you may be asked to approve the redrawn figure before your paper is accepted.

-Please upload separate high quality figure files via the submission form.

-Author profile(s) must be uploaded via the submission form. Authors should submit a short biography (no more than 100 words for one author or 150 words in total for two authors) and a portrait photograph of the two leading authors on the paper. These should be uploaded, clearly labelled, with the manuscript submission. Any standard image format for the photograph is acceptable, but the resolution should be at least 300 dpi and preferably more. A group photograph of all authors is also acceptable, providing the biography for the whole group does not exceed 150 words.

EDITOR COMMENTS

Reviewing Editor:

Thank you for writing this Review article. Reviewers found it to be clearly written and generally good, but in need of some revision as described. To summarize, the manuscript would benefit from more molecular mechanisms, inclusion of more recent literature, and perhaps some additional figures.

REFEREE COMMENTS

Referee #1:

This commissioned review summarizes literature on the molecular mechanisms governing dysregulation of Ca and Na in heart failure. It is well organized, a pleasure to read, and provides scholarly references of the history of the field through current thinking. Unsettled controversies are also explained. This reference will be useful for trainees in the field of heart failure and for scientists and professionals in adjacent fields seeking an efficient way to broaden their knowledge.

Major comments:

1. The review could be a bit more up-to-date. In the most recent four publication years including 2019 there are only 5 references, and from a search in PubMed there seem to be several relevant publications in the last two years that could reasonably be included to make the review more timely.

2. The abstract and introduction mention adaptations involving sympathetic stimulation, release of natriuretic peptides, and activation of the renin-angiotensin-aldosterone pathway. It is not clear how these adaptations are linked to changes in Ca and Na regulation that are the focus of the review. Making that link, however briefly, would improve the manuscript.

Minor comments:

1. Suggest referring at least once in the paragraph "Na/Ca exchange" to "NCX," especially since NCX is used in Fig. 1.
2. In first full paragraph on p. 5, the sentence beginning with "Peak values of Ca" is a bit cumbersome and readability could be improved with a rewrite.
3. Also on p. 5, the final sentence of the second paragraph could be edited for clarity.
4. The next sentence, at the beginning of the third paragraph on p. 5, is a bit dense and could also be edited for readability.

Referee #2:

This is a clearly written and concise review by dr MacLeod on sodium and calcium regulation during heart failure. I do have some suggestions for further improvement of the review.

For easier indication of the suggested changes and additions, I have labeled the first page containing the abstract page 1, and so forth.

Page 3: "Arrhythmia in heart failure", here it is mentioned "a disturbance of rhythm" but this should be more clarified and specified (brady-arrhythmia, tachycardia, sudden arrhythmic death?)

Page 3 "to the production of contraction" I would remove "the production of"

Pages 3-4: in the discussion on calcium influx, I would also include enhanced late sodium current; overall, I find this issue underrepresented in the review; for instance, the impact of CamKII on late sodium current is not mentioned, and also there is no mention of the role of "neuronal" sodium channels which have been shown to contribute to ionic changes during heart failure

Page 4: PKA and CamKII are mentioned briefly here, but I think they deserve more detailed discussion; overall, changes in post-translational regulation of e.g. ion channels and transporters in the setting of HF could be addressed in more detail

Pages 5-6: CamKII is mentioned on page 5, but later on page 6 in more detail. I would change the order and introduce CamKII (the information on page 6) earlier on (see also previous point). Could you also speculate on what changes first: CamKII or calcium homeostasis?

Page 6: on a general note, when you speak of heart failure, is there a unified mechanism for all types of heart failure, or does it depend on the cause of HF?

Page 7: there is quite a long section on the clinical effects of digoxin; while this is interesting, it does seem a bit unbalanced since this is not done in so much detail for other (potential) therapeutic approaches.

Page 7, last paragraph: I would remove this here since it is addressed also on page 9 and hence it is repetitive.

Page 8, first 4 sentences ("Lastly, digoxin increase..."): this does not fit here and appears to part of the previous section on digoxin? However, please also note my previous comment on the clinical digoxin section.

Page 8 "However, in a rabbit model of heart failure a decrease in both..." this sentence and the following one are confusing, I would suggest to rewrite.

Page 9: the section on "Na⁺ influx in heart failure" would benefit from a more detailed discussion on late sodium current, including the role of CamKII, post-translational regulation, and potential impact of neuronal sodium channels (non-Nav1.5; see for instance PMID: 29931291). There are quite some more recent papers that have investigated this. Also, the sentence on the clinical trial with ranolazine and arrhythmias needs expanding on (or removing), since there have been more studies since then and there are also potential pro-arrhythmic side-effects of the drug.

Pages 9-10: the impact of ionic changes on mitochondrial function is certainly relevant, but the other way around is likely also important; metabolic changes during heart failure may be associated with for instance increased ROS production which in turn may affect (among others) CamKII, calcium and (late) sodium. This aspect is not clear in the current description in the review.

General remarks:

While I understand that it is sometimes difficult to distinguish "the chicken from the egg", it would be valuable if you could include some (hypothetical) considerations on the potential sequence of events. Does metabolic/mitochondrial dysfunction in HF precede alterations in ionic homeostasis? Do changes in CamKII precede calcium dysregulation or vice versa?

Overall, the review contains somewhat limited information on molecular mechanisms and signaling pathways (see also comments on PKA, CamKII, etc). Would it be possible to expand on this? An informative figure on this would also be very helpful.

I noticed that the vast majority of cited references (in particular, those of original experimental work) have been published over 10 years ago. It would be good to include some more recent experimental studies, in particular those on (molecular) mechanisms. In addition, recent work on SGLT2 inhibitors should be discussed.

Nomenclature: gene/mRNA names should be in italics (human all in capitals, mouse only the first letter in capital). Some of the ion currents are indicated incorrectly, for instance in "Nav1.5", only "V" should be in subscript.

END OF COMMENTS

Confidential Review

16-May-2022

JP-SR-2022-283082

Responses to Reviewers

I thank the Senior Editor and the Reviewers for their pertinent and helpful comments. I have taken these on board, added some more detail to the text, corrected and clarified existing text where necessary and updated references. I think the new version benefits from these inclusions and refinements and hope it is now satisfactory.

Referee #1:

This commissioned review summarizes literature on the molecular mechanisms governing dysregulation of Ca and Na in heart failure. It is well organized, a pleasure to read, and provides scholarly references of the history of the field through current thinking. Unsettled controversies are also explained. This reference will be useful for trainees in the field of heart failure and for scientists and professionals in adjacent fields seeking an efficient way to broaden their knowledge.

Many thanks

Major comments:

1. The review could be a bit more up-to-date. In the most recent four publication years including 2019 there are only 5 references, and from a search in PubMed there seem to be several relevant publications in the last two years that could reasonably be included to make the review more timely.

This was appreciated. In trying to include some historical perspective so that the reader could understand the development of ideas, the balance between old and new swung too much to the former. This has been rectified with the inclusion of CaMKII effects, some more complex aspects of late Na current and the puzzling action of SGLT2 inhibitors.

2. The abstract and introduction mention adaptations involving sympathetic stimulation, release of natriuretic peptides, and activation of the renin-angiotensin-aldosterone pathway. It is not clear how these adaptations are linked to changes in Ca and Na regulation that are the focus of the review. Making that link, however briefly, would improve the manuscript.

Some more explanation has been added to the Introduction section and the link made between pressure and/or volume overload caused by SNS and RAAS activation and changes in cardiac cell biology. This was implied before but is now more distinct.

Minor comments:

1. Suggest referring at least once in the paragraph "Na/Ca exchange" to "NCX," especially since NCX is used in Fig. 1.

NCX has replaced Na⁺/Ca²⁺ exchange following its initial use.

2. In first full paragraph on p. 5, the sentence beginning with "Peak values of Ca" is a bit cumbersome and readability could be improved with a rewrite.

Rewritten

3. Also on p. 5, the final sentence of the second paragraph could be edited for clarity.

Rewritten

4. The next sentence, at the beginning of the third paragraph on p. 5, is a bit dense and could also be edited for readability.

Rewritten

Referee #2:

This is a clearly written and concise review by dr MacLeod on sodium and calcium regulation during heart failure. I do have some suggestions for further improvement of the review.

Many thanks

For easier indication of the suggested changes and additions, I have labeled the first page containing the abstract page 1, and so forth.

Page 3: "Arrhythmia in heart failure", here it is mentioned "a disturbance of rhythm" but this should be more clarified and specified (brady-arrhythmia, tachycardia, sudden arrhythmic death?)

The problem is that no one form of dysrhythmia predominates hence this is best left more generalised.

Page 3 "to the production of contraction" I would remove "the production of"

Done

Pages 3-4: in the discussion on calcium influx, I would also include enhanced late sodium current; overall, I find this issue underrepresented in the review; for instance, the impact of CamKII on late sodium current is not mentioned, and also there is no mention of the role of "neuronal" sodium channels which have been shown to contribute to ionic changes during heart failure

Page 4: PKA and CamKII are mentioned briefly here, but I think they deserve more detailed discussion; overall, changes in post-translational regulation of e.g. ion channels and transporters in the setting of HF could be addressed in more detail

Pages 5-6: CamKII is mentioned on page 5, but later on page 6 in more detail. I would change the order and introduce CamKII (the information on page 6) earlier on (see also previous point). Could you also speculate on what changes first: CamKII or calcium homeostasis?

Many thanks to the reviewer in picking up this omission. The above three comments have been addressed. There is now a paragraph on CaMKII and later in the article reference to its effects on late Na⁺ current.

Page 6: on a general note, when you speak of heart failure, is there a unified mechanism for all types of heart failure, or does it depend on the cause of HF?

Cardiologists have been searching for a suitable definition of heart failure for many years. Relatively simple definitions (Harris, 1994. PMID: 7947360) become more debated (Tan, 2010. PMID: 20136608) and more encompassing as knowledge improves (Ponikowski, 2016. PMID: 27206819) and then start to include some clinically measurable indices (Bozkurt, 2021. PMID: 33662581). Most would agree with the Ponikowski et al., definition; "a clinical syndrome characterized by typical signs and symptoms, caused by a structural and/or functional cardiac abnormality, resulting in a reduced cardiac output and/or elevated intracardiac pressures at rest or during stress," and, in answer to your question, I would add that it is the end result of many quite distinct and varied disease states. This is what I mean when I speak of heart failure.

At present a unifying mechanism is lacking probably due to the (1) variation in insults that seem to lead to its evolution and (2) complex feedback loops that are established to compensate for it.

Page 7: there is quite a long section on the clinical effects of digoxin; while this is interesting, it does seem a bit unbalanced since this is not done in so much detail for other (potential) therapeutic approaches.

This section has been shortened.

Page 7, last paragraph: I would remove this here since it is addressed also on page 9 and hence it is repetitive.

Removed

Page 8, first 4 sentences ("Lastly, digoxin increase..."): this does not fit here and appears to part of the previous section on digoxin? However, please also note my previous comment on the clinical digoxin section.

This section has been tidied up.

Page 8 "However, in a rabbit model of heart failure a decrease in both..." this sentence and the following one are confusing, I would suggest to rewrite.

This has been rewritten.

Page 9: the section on "Na⁺ influx in heart failure" would benefit from a more detailed discussion on late sodium current, including the role of CamKII, post-translational regulation, and potential impact of neuronal sodium channels (non-Nav1.5; see for instance PMID: 29931291). There are quite some more recent papers that have investigated this. Also, the sentence on the clinical trial with ranolazine and arrhythmias needs expanding on (or removing), since there have been more studies since then and there are also potential pro-arrhythmic side-effects of the drug.

More information has been given on the role of CaMKII and late Na⁺ current and the sentence about ranolazine removed. I agree with the Reviewer that more recent evidence points to complex effects of ranolazine which are too multifaceted to discuss succinctly.

Pages 9-10: the impact of ionic changes on mitochondrial function is certainly relevant, but the other way around is likely also important; metabolic changes during heart failure may be associated with for instance increased ROS production which in turn may affect (among others) CamKII, calcium and (late) sodium. This aspect is not clear in the current description in the review.

I fully agree that ROS production will affect the mitochondria and cell metabolism, in turn affecting a number of processes. The idea of the review was to illustrate that alterations to cellular Na⁺ and Ca²⁺ regulation influence and promote a spectrum of pathophysiological changes that eventually become maladaptive. However, I accept this is perhaps too much of a "one-way" perspective. I have added a small section here to improve the balanceI have now tried to tie in increased ROS with angiotensin II and introduce the idea that increased mitochondrial Ca concentration also disturbs function.

General remarks:

While I understand that it is sometimes difficult to distinguish "the chicken from the egg", it would be valuable if you could include some (hypothetical) considerations on the potential sequence of events. Does metabolic/mitochondrial dysfunction in HF precede alterations in ionic homeostasis? Do changes in CamKII precede calcium dysregulation or vice versa?

I am not sure if the responses can be "sequenced" in this way. Not all hypertrophic responses are maladaptive. Exercise-induced (or pregnancy-induced) cardiac remodelling takes place and (usually) there is no subsequent decompensation so pinpointing causation (ie X precedes Y therefore X is the cause) is difficult. Presumably in the initial phases, the compensatory responses taking place in physiological and pathological remodelling overlap which tends to suggest complex interplay. Dorn (PMID: 17389260) suggests that very early signalling events are different with some potential for crossover. Given most biological systems seem to have large amounts of redundancy I think there could be many points for interaction.

We now understand quite a lot about myocardial disease phenotypes at various timepoints (because specific timepoints are relatively easy to study) but we know much less about

disease progression (ie longitudinal studies of compensation and decompensation phases). In these phases multiple pathways are likely to be simultaneously activated making determining the chicken and egg challenging.

Overall, the review contains somewhat limited information on molecular mechanisms and signaling pathways (see also comments on PKA, CamKII, etc). Would it be possible to expand on this? An informative figure on this would also be very helpful.

I have included another figure on the likely CaMKII signalling pathways.

I noticed that the vast majority of cited references (in particular, those of original experimental work) have been published over 10 years ago. It would be good to include some more recent experimental studies, in particular those on (molecular) mechanisms. In addition, recent work on SGLT2 inhibitors should be discussed.

This was not deliberate but simply arises from referencing foundation work. Hopefully the criticism has been rectified with the inclusion of CaMKII effects, some more complex aspects of late Na current and the puzzling action of SGLT2 inhibitors.

Nomenclature: gene/mRNA names should be in italics (human all in capitals, mouse only the first letter in capital). Some of the ion currents are indicated incorrectly, for instance in "Nav1.5", only "V" should be in subscript.

Corrected

Dear Professor MacLeod,

Re: JP-SR-2022-283082R1 "Changes in cellular Ca²⁺ and Na⁺ regulation during the progression towards heart failure" by Kenneth T. MacLeod

I am pleased to tell you that your Symposium Review article has been accepted for publication in The Journal of Physiology, subject to any modifications to the text that may be required by the Journal Office to conform to House rules.

NEW POLICY: In order to improve the transparency of its peer review process The Journal of Physiology publishes online as supporting information the peer review history of all articles accepted for publication. Readers will have access to decision letters, including all Editors' comments and referee reports, for each version of the manuscript and any author responses to peer review comments. Referees can decide whether or not they wish to be named on the peer review history document.

The last Word version of the paper submitted will be used by the Production Editors to prepare your proof. When this is ready you will receive an email containing a link to Wiley's Online Proofing System. The proof should be checked and corrected as quickly as possible.

All queries at proof stage should be sent to tjp@wiley.com.

The accepted version of the manuscript is the version that will be published online until the copy edited and typeset version is available. Authors should note that it is too late at this point to offer corrections prior to proofing. Major corrections at proof stage, such as changes to figures, will be referred to the Reviewing Editor for approval before they can be incorporated. Only minor changes, such as to style and consistency, should be made a proof stage. Changes that need to be made after proof stage will usually require a formal correction notice.

Are you on Twitter? Once your paper is online, why not share your achievement with your followers. Please tag The Journal (@jphysiol) in any tweets and we will share your accepted paper with our 22,000+ followers!

Yours sincerely,

Professor Laura Bennet
Senior Editor
The Journal of Physiology
<https://jp.msubmit.net>
<http://jp.physoc.org>
The Physiological Society
Hodgkin Huxley House
30 Farringdon Lane
London, EC1R 3AW
UK
<http://www.physoc.org>
<http://journals.physoc.org>

EDITOR COMMENTS:

Reviewing Editor:

Thank you for making significant revisions in response to the reviewers' comments. The revised work is improved, and likely to be impactful for the field of heart failure and ion channels more generally.

REFeree COMMENTS:

Referee #1:

My concerns have been addressed and the manuscript has been improved by the consideration of all reviewers' comments.

Referee #2:

The author has more than adequately addressed my comments and suggestions, and has made significant changes to the manuscript which is now further improved. I would like to congratulate the author on an excellent review.

*** IMPORTANT NOTICE ABOUT OPEN ACCESS ***

To assist authors whose funding agencies mandate public access to published research findings sooner than 12 months after publication The Journal of Physiology allows authors to pay an open access (OA) fee to have their papers made freely available immediately on publication.

You will receive an email from Wiley with details on how to register or log-in to Wiley Authors Services where you will be able to place an OnlineOpen order.

You can check if your funder or institution has a Wiley Open Access Account here <https://authorservices.wiley.com/author-resources/Journal-Authors/licensing-and-open-access/open-access/author-compliance-tool.html>

Your article will be made Open Access upon publication, or as soon as payment is received.

If you wish to put your paper on an OA website such as PMC or UKPMC or your institutional repository within 12 months of publication you must pay the open access fee, which covers the cost of publication.

OnlineOpen articles are deposited in PubMed Central (PMC) and PMC mirror sites. Authors of OnlineOpen articles are permitted to post the final, published PDF of their article on a website, institutional repository, or other free public server, immediately on publication.

Note to NIH-funded authors: The Journal of Physiology is published on PMC 12 months after publication, NIH-funded authors DO NOT NEED to pay to publish and DO NOT NEED to post their accepted papers on PMC.

1st Confidential Review

18-Jul-2022